# G2SAT: Learning to Generate SAT Formulas

**Jiaxuan You**[1]*
jiaxuan@cs.stanford.edu

**Haoze Wu**[1]*
haozewu@stanford.edu

**Clark Barrett**[1]
barrett@cs.stanford.edu

**Raghuram Ramanujan**[2]
raramanujan@davidson.edu

**Jure Leskovec**[1]
jure@cs.stanford.edu

[1]Department of Computer Science, Stanford University
[2]Department of Mathematics and Computer Science, Davidson College

## Abstract

The Boolean Satisfiability (SAT) problem is the canonical NP-complete problem and is fundamental to computer science, with a wide array of applications in planning, verification, and theorem proving. Developing and evaluating practical SAT solvers relies on extensive empirical testing on a set of real-world benchmark formulas. However, the availability of such real-world SAT formulas is limited. While these benchmark formulas can be augmented with synthetically generated ones, existing approaches for doing so are heavily hand-crafted and fail to simultaneously capture a wide range of characteristics exhibited by real-world SAT instances. In this work, we present G2SAT, the first deep generative framework that learns to generate SAT formulas from a given set of input formulas. Our key insight is that SAT formulas can be transformed into latent bipartite graph representations which we model using a specialized deep generative neural network. We show that G2SAT can generate SAT formulas that closely resemble given real-world SAT instances, as measured by both graph metrics and SAT solver behavior. Further, we show that our synthetic SAT formulas could be used to improve SAT solver performance on real-world benchmarks, which opens up new opportunities for the continued development of SAT solvers and a deeper understanding of their performance.

## 1 Introduction

The *Boolean Satisfiability (SAT) problem* is central to computer science, and finds many applications across Artificial Intelligence, including planning [24], verification [7], and theorem proving [14]. SAT was the first problem to be shown to be NP-complete [9], and there is believed to be no general procedure for solving arbitrary SAT instances efficiently. Nevertheless, modern solvers are able to routinely decide large SAT instances in practice, with different algorithms proving to be more successful than others on particular problem instances. For example, incomplete search methods such as WalkSAT [35] and survey propagation [6] are more effective at solving large, randomly generated formulas, while complete solvers leveraging *conflict-driven clause learning* (CDCL) [30] fare better on large structured SAT formulas that commonly arise in industrial settings.

Understanding, developing and evaluating modern SAT solvers relies heavily on extensive empirical testing on a suite of benchmark SAT formulas. Unfortunately, in many domains, availability of

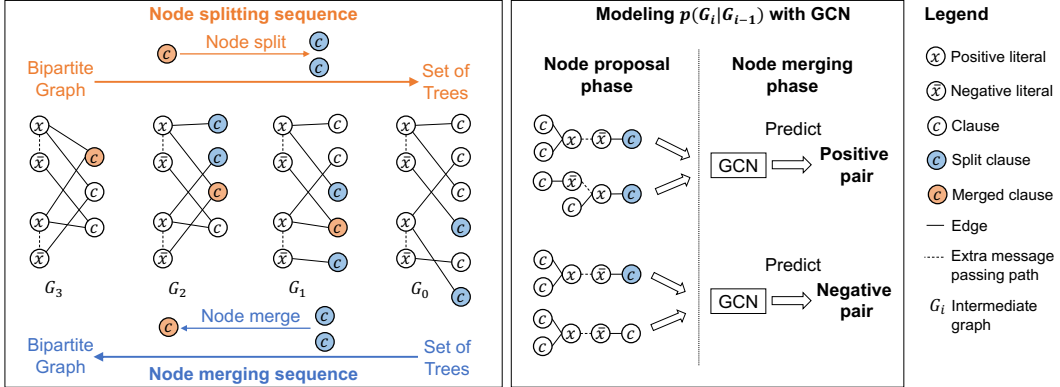

Figure 1: An overview of the proposed G2SAT model. **Top left**: A given bipartite graph can be decomposed into a set of disjoint trees by applying a sequence of node splitting operations. Orange node $c$ in graph $G_i$ is split into two blue $c$ nodes in graph $G_{i-1}$. Every time a node is split, one more node appears in the right partition. **Right**: We use node pairs gathered from such a sequence of node splitting operations to train a GCN-based classifier that predicts whether a pair of $c$ nodes should be merged. **Bottom left**: Given such a classifier, G2SAT generates a bipartite graph by starting with a set of trees $G_0$ and applying a sequence of node merging operations, where two blue nodes in graph $G_{i-1}$ get merged in graph $G_i$. G2SAT uses the GCN-based classifier that captures the bipartite graph structure to sequentially decide which nodes to merge from a set of candidates. Best viewed in color.

benchmarks is still limited. While this situation has improved over the years, new and interesting benchmarks — both real and synthetic — are still in demand and highly welcomed by the SAT community. Developing expressive generators of structured SAT formulas is important, as it would provide for a richer set of evaluation benchmarks, which would in turn allow for the development of better and faster SAT solvers. Indeed, the problem of pseudo-industrial SAT formula generation has been identified as one of the ten key challenges in propositional reasoning and search [36].

One promising approach for tackling this challenge is to represent SAT formulas as graphs, thus recasting the original problem as a graph generation task. Specifically, every SAT formula can be converted into a corresponding bipartite graph, known as its literal-clause graph (LCG), via a bijective mapping. Prior work in pseudo-industrial SAT instance generation has relied on hand-crafted algorithms [15, 16], focusing on capturing one or two of the graph statistics exhibited by real-world SAT formulas [1, 34]. As researchers continue to uncover new and interesting characteristics of real-world SAT instances [1, 23, 34, 31], previous SAT generators might become invalid, and hand-crafting new models that simultaneously capture all the pertinent properties becomes increasingly difficult. On the other hand, recent work on deep generative models for graphs [4, 29, 43, 45] has demonstrated their ability to capture *many* of the essential features of real-world graphs such as social networks and citation networks, as well as graphs arising in biology and chemistry. However, these models do not enforce bipartiteness and therefore cannot be directly employed in our setting. While it might be possible to post-process these generated graphs, such a solution would be ad hoc, computationally expensive and would fail to exploit the unique structure of bipartite graphs.

In this paper, we present G2SAT, the first deep generative model that *learns* to generate SAT formulas based on a given set of input formulas. We use LCGs to represent SAT formulas, and formulate the task of SAT formula generation as a bipartite graph generation problem. Our key insight is that any bipartite graph can be generated by starting with a set of trees, and then applying a sequence of *node merging* operations over the nodes from one of the two partitions. As we merge nodes, trees are also merged, and complex bipartite structures begin to appear (Figure 1, left). In this manner, a set of input bipartite graphs (SAT formulas) can be characterized by a distribution over the sequence of node merging operations. Assuming we can capture/learn the distribution over the pairs of nodes to merge, we can start with a set of trees and then keep merging nodes in order to generate realistic bipartite graphs (i.e., realistic SAT formulas). G2SAT models this iterative node merging process in an auto-regressive manner, where a node merging operation is viewed as a sample from the underlying conditional distribution that is parameterized by a Graph Convolutional Neural Network (GCN) [18, 19, 27, 44], and the same GCN is shared across all the steps of the generation process.

This formulation raises the following question: how do we devise a sequential generative process when we are only given a static input SAT formula? In other words, how do we generate training data for our generator? We resolve this challenge as follows (Figure 1). We define *node splitting* as the inverse operation to node merging. We apply this node splitting operation to a given input bipartite graph (a real-world SAT formula) and decompose it into a set of trees. We then reverse the splitting sequence, so that we start with a set of trees and learn from the sequence of node merging operations that recovers a realistic SAT formula. We train a GCN-based classifier that decides which two nodes to merge next, based on the structure of the graph generated so far.

At graph generation time, we initialize G2SAT with a set of trees and iteratively merge node pairs based on the conditional distribution parameterized by the trained GCN model, until a user-specified stopping criterion is met. We utilize an efficient two-phase generation procedure: in the node proposal phase, candidate node pairs are randomly drawn, and in the node merging phase, the learned GCN model is applied to select the most likely node pair to merge.

Experiments demonstrate that G2SAT is able to generate formulas that closely resemble the input real-world SAT instances in many graph-theoretic properties such as modularity and the presence of scale-free structures, with $24\%$ lower relative error compared to state-of-the-art methods. More importantly, G2SAT generates formulas that exhibit the same hardness characteristics as real-world SAT formulas in the following sense: when the generated instances are solved using various SAT solvers, those solvers that are known to be effective on structured, real-world instances consistently outperform those solvers that are specialized in solving random SAT instances. Moreover, our results suggest that we can use our synthetically generated formulas to more effectively tune the hyperparameters of SAT solvers, achieving an $18\%$ speedup in run time on unseen formulas, compared to tuning the solvers only on the formulas used for training.[1]

## 2 Preliminaries

**Goal of generating SAT formulas**. Our goal is to design a SAT generator that, given a suite of SAT formulas, generates new SAT formulas with similar properties. Our aim is to capture not only graph theoretic properties, but also realistic SAT solver behavior. For example, if we train our G2SAT model on formulas from application domain $X$, then solvers that traditionally excel in solving problems in domain $X$, should also excel in solving synthetic G2SAT formulas (rather than, say, solvers that specialize in solving random SAT formulas).

**SAT formulas and their graph representations**. A SAT formula $\phi$ is composed of Boolean variables $x_i$ connected by the logical operators *and* ($\land$), *or* ($\lor$), and *not* ($\neg$). A formula is satisfiable if there exists an assignment of Boolean values to the variables such that the overall formula evaluates to true. In this paper, we are concerned with formulas in Conjunctive Normal Form (CNF)[2], i.e., formulas expressed as conjunctions of disjunctions. Each disjunction is called a *clause*, while a Boolean variable $x_i$ or its negation $\neg x_i$ is called a *literal*. For example, $(x_1 \lor x_2 \lor \neg x_3) \land (\neg x_1 \lor \neg x_2)$ is a CNF formula with two clauses that can be satisfied by assigning true to $x_1$ and false to $x_2$.

Traditionally, researchers have studied four different graph representations for SAT formulas [3]: (1) *Literal-Clause Graph (LCG)*: there is a node for each literal and each clause, with an edge denoting the occurrence of a literal in a clause. An LCG is bipartite and there exists a bijection between CNF formulas and LCGs. (2) *Literal-Incidence Graph (LIG)*: there is a node for each literal and two literals have an edge if they co-occur in a clause. (3) *Variable-Clause Graph (VCG)*: obtained by merging the positive and negative literals of the same variables in an LCG. (4) *Variable-Incidence Graph (VIG)*: obtained by performing the same literal merging operation on the LIG. In this paper, we use LCGs to represent SAT formulas.

**LCGs as bipartite graphs**. We represent a bipartite graph $G = (\mathcal{V}^G, \mathcal{E}^G)$ by its node set $\mathcal{V}^G = \{v_1^G, ..., v_n^G\}$ and edge set $\mathcal{E}^G \subseteq \{(v_i^G, v_j^G) | v_i^G, v_j^G \in \mathcal{V}^G\}$. In the rest of paper, we omit the superscript $G$ whenever it is possible to do so. Nodes in a bipartite graph can be split into two disjoint partitions $\mathcal{V}_1$ and $\mathcal{V}_2$ such that $\mathcal{V} = \mathcal{V}_1 \cup \mathcal{V}_2$. Edges only exist between nodes in different partitions, i.e., $\mathcal{E} \subseteq \{(v_i, v_j) | v_i \in \mathcal{V}_1, v_j \in \mathcal{V}_2\}$. An LCG with $n$ literals and $m$ clauses has $\mathcal{V}_1 = \{l_1, ..., l_n\}$

and $\mathcal{V}_2 = \{c_1, ..., c_m\}$, where $\mathcal{V}_1$ and $\mathcal{V}_2$ are referred to as the literal partition and the clause partition, respectively. We may also write out $l_i$ as $x_i$ or $\neg x_i$ when specifying the literal sign is necessary.

**Benefits of using LCGs to generate SAT formulas**. While we choose to work with LCGs because they are bijective to SAT formulas, the LIG is also a viable alternative. Unlike LCGs, there are no explicit constraints over LIGs, and thus, previously developed general deep graph generators could in principle be used. However, the ease of generating LIGs is offset by the fact that key information is lost during the translation from the corresponding SAT formula. In particular, given a pair of literals, the LIG only encodes whether they co-occur in a clause but fails to capture how many times and in which clauses they co-occur. It can further be shown that an LIG corresponds to a number of SAT formulas that is at least exponential in the number of 3-cliques in the LIG. This ambiguity severely limits the usefulness of LIGs for SAT benchmark generation.

## 3   Related Work

**SAT Generators**. Existing synthetic SAT generators are hand-crafted models that are typically designed to generate formulas that fit a particular graph statistic. The mainstream generators for pseudo-industrial SAT instances include the Community Attachment (CA) model [15], which generates formulas with a given VIG modularity, and the Popularity-Similarity (PS) model [16], which generates formulas with a specific VCG degree distribution. In addition, there are also generators for random $k$-SAT instances [5] and crafted instances that come from translations of structured combinatorial problems, such as graph coloring, graph isomorphism, and Ramsey numbers [28]. Currently, all SAT generators are hand-crafted and machine learning provides an exciting alternative.

**Deep Graph Generators**. Existing deep generative models of graphs fall into two categories. In the first class are models that focus on generating perturbations of a given graph, by direct decoding from computed node embeddings [26] or latent variables [17], or learning the random walk distribution of a graph [4]. The second class comprises models that can learn to generate a graph by sequentially adding nodes and edges [29, 45, 43]. Domain specific generators for molecular graphs [10, 22] and 3D point cloud graphs [40] have also been developed. However, current deep generative models of graphs do not readily apply to SAT instance generation. Thus, we develop a novel bipartite graph generator that respects all the constraints imposed by graphical representations of SAT formulas and generates the formula graph via a sequence of node merging operations.

**Deep learning for SAT**. NeuroSAT also represents SAT formulas as graphs and computes node embeddings using GCNs [38]. However, NeuroSAT focuses on using the embeddings to solve SAT formulas, while we aim to generate SAT formulas. A preliminary version of the work presented in this paper appeared in [41], where existing graph generative models were used to learn the LIG of a SAT formula. However, extensive post-processing was required to extract a formula from a generated LIG, since an LIG is an ambiguous representation of a SAT formula. In this work, we develop a new deep graph generative model, that, unlike existing techniques, is able to directly learn the bijective graph representation of a SAT formula, and therefore better capture its characteristics.

## 4   The G2SAT Framework

### 4.1   G2SAT: Generating Bipartite Graphs by Iterative Node Merging Operations

As discussed in Section 2, a SAT formula is uniquely represented by its LCG which is a bipartite graph. From the perspective of generative models, our primary objective is to learn a distribution over bipartite graphs, based on a set of observed bipartite graphs $\mathbb{G}$ sampled from the data distribution $p(G)$. Each bipartite graph $G \in \mathbb{G}$ may have a different number of nodes and edges. Due to the complex dependency between nodes and edges, directly learning $p(G)$ is challenging. Therefore, we generate a graph via an $n$-step iterative process, $p(G) = \prod_{i=1}^{n} p(G_i|G_1, ..., G_{i-1})$, where $G_i$ refers to an intermediate graph at step $i$ in the iterative generation process. Since we focus on generating static graphs, we assume that the order of the generation trajectory does not matter, as long as the same graph is generated. This assumption implies the following Markov property over the conditional distribution, $p(G_i|G_1, ..., G_{i-1}) = p(G_i|G_{i-1})$.

The key to a successful iterative graph generative model is a proper instantiation of the conditional distribution $p(G_i|G_{i-1})$. Existing approaches [29, 43, 45] often model $p(G_i|G_{i-1})$ as the distribution

over the random addition of nodes or edges to $G_{i-1}$. While in theory this formulation allows the generation of any kind of graph, it cannot satisfy the hard partition constraint for bipartite graphs. In contrast, our proposed G2SAT has a simple generation process that is guaranteed to preserve the bipartite partition constraint, without the need for hand-crafted generation rules or post-processing procedures. The G2SAT framework relies on node splitting and merging operations, which are defined as follows.

**Definition 1.** *The node splitting operation, when applied to node $v$, removes some edges between $v$ and its neighboring nodes, and then connects those edges to a new node $u$. The node merging operation, when performed over two nodes $u$ and $v$, removes all the edges between $v$ and its neighboring nodes, and then connects those edges to $u$. Formally,* $\mathrm{NodeSplit}(u, G)$ *returns a tuple* $(u, v, G')$, *and* $\mathrm{NodeMerge}(u, v, G)$ *returns a tuple* $(u, G')$.

Note that according to this definition, a node merging operation can always be reversed by a node splitting operation. The core idea underlying G2SAT is then motivated by the following observation.

**Observation 1.** *A bipartite graph can always be transformed into a set of trees by a sequence of node splitting operations over the nodes in one of the partitions.*

The proof of this claim follows from the fact that the node splitting operation strictly reduces a node's degree. Therefore, repeatedly applying node splitting to all the nodes in a partition will ultimately reduce the degree of all those nodes to 1, producing a set of trees (Figure 1, Left). This observation implies that a bipartite graph can always be generated via *a sequence of node merging operations*. In G2SAT, we always merge clause nodes in the clause partition $\mathcal{V}_2^{G_{i-1}}$ for a given graph $G_{i-1}$. We then design the following instantiation of $p(G_i|G_{i-1})$,

$$p(G_i|G_{i-1}) = p(\mathrm{NodeMerge}(u, v, G_{i-1})|G_{i-1}) = \mathrm{Multimonial}(\mathbf{h}_u^T \mathbf{h}_v / Z | \forall u, v \in \mathcal{V}_2^{G_{i-1}}) \quad (1)$$

where $\mathbf{h}_u$ and $\mathbf{h}_v$ are the embeddings for nodes $u$ and $v$, and $Z$ is the normalizing constant that ensures that the distribution $\mathrm{Multimonial}(\cdot)$ is valid. We aim for embeddings $\mathbf{h}_u$ that capture the multi-hop neighborhood structure of a node $u$ and that can be computed from a single trainable model. Further, this model needs to be capable of generalizing across different generation stages and different graphs. Therefore, we use the GraphSAGE framework [18] to compute node embeddings, which is a variant of GCNs that has been shown to have strong inductive learning capabilities across different graphs. Specifically, the $l$-th layer of GraphSAGE can be written as

$$\mathbf{n}_u^{(l)} = \mathrm{AGG}(\mathrm{RELU}(\mathbf{Q}^{(l)}\mathbf{h}_v^{(l)} + \mathbf{q}^{(l)}|v \in N(u)))$$
$$\mathbf{h}_u^{(l+1)} = \mathrm{RELU}(\mathbf{W}^{(l)}\mathrm{CONCAT}(\mathbf{h}_u^{(l)}, \mathbf{n}_u^{(l)}))$$

where $\mathbf{h}_u^{(l)}$ is the $l$-th layer node embedding for node $u$, $N(u)$ is the local neighborhood of $u$, $\mathrm{AGG}(\cdot)$ is an aggregation function such as mean pooling, and $\mathbf{Q}^{(l)}, \mathbf{q}^{(l)}, \mathbf{W}^{(l)}$ are trainable parameters. The input node features are length-3 one-hot vectors, which are used to represent the three node types in LCGs, i.e., positive literals, negative literals and clauses. In addition, since each literal and its negation are closely related, we add an additional message passing path between them.

## 4.2 Scalable G2SAT with Two-phase Generation Scheme

LCGs can easily have tens of thousands of nodes; thus, there are millions of candidate node pairs that could be merged. This makes the computation of the normalizing constant $Z$ in Equation 1 infeasible. To avoid this issue, we design a two-phase scheme to instantiate Equation 1, which includes a node proposal phase and a node merging phase (Figure 1, right). Intuitively, the idea is to begin with a fixed oracle that proposes random candidate node pairs. Then, a model only needs to decide if the proposed node pair should be merged or not, which is an easier learning task compared to selecting from among millions of candidate options. Instead of directly learning and sampling from $p(G_i|G_{i-1})$, we introduce additional random variables $u$ and $v$ to represent random nodes, and then learn the joint distribution $p(G_i, u, v|G_{i-1}) = p(u, v|G_{i-1})p(G_i|G_{i-1}, u, v)$. Here, $p(u, v|G_{i-1})$ corresponds to the node proposal phase and $p(G_i|G_{i-1}, u, v)$ models the node merging phase.

In theory, $p(u, v|G_{i-1})$ can be any distribution as long as it has non-empty support. Since LCGs are inherently static graphs, there is little prior knowledge or additional information on how this iterative generation process should proceed. Therefore, we implement the node proposal phase such that a

random node pair is sampled from all candidate clause nodes uniformly at random. Then, in the node merging phase, instead of computing the dot product between all possible node pairs, the model only needs to compute the dot product between the sampled node pairs. Specifically, we have

$$p(G_i, u, v | G_{i-1}) = p(u, v | G_{i-1}) p(\text{NodeMerge}(u, v, G_{i-1}) | G_{i-1}, u, v)$$
$$= \text{Uniform}(\{(u, v) | \forall u, v \in \mathcal{V}_2^{G_{i-1}}\}) \, \text{Bernoulli}(\sigma(\mathbf{h}_u^T \mathbf{h}_v) | u, v) \quad (2)$$

where Uniform is the discrete uniform distribution and $\sigma(\cdot)$ is the sigmoid function.

### 4.3 G2SAT at Training Time

The two-phase generation scheme described in Section 4.2 transforms the bipartite graph generation task into a binary classification task. We train the classifier to minimize the following binary cross entropy loss:

$$\mathcal{L} = -\mathbb{E}_{u, v \sim p_{pos}}[\log(\sigma(\mathbf{h}_u^T \mathbf{h}_v))] - \mathbb{E}_{u, v \sim p_{neg}}[\log(1 - \sigma(\mathbf{h}_u^T \mathbf{h}_v))] \quad (3)$$

where $p_{pos}$ and $p_{neg}$ are the distributions over positive and negative training examples (i.e. node pairs). We say a node pair is a positive example if the node pair should be merged according to the training set. To acquire the necessary training data from input bipartite graphs, we develop a procedure that is described in Algorithm 1. Given an input bipartite graph $G$, we apply the node splitting operation to the graph for $n = |\mathcal{E}^G| - |\mathcal{V}_2^G|$ steps, which guarantees that the input graph will be decomposed into a set of trees. Within each step, a random node $s$ in partition $\mathcal{V}_2^{G_i}$ with degree greater than 1 is chosen for splitting, and a random subset of edges that connect to $s$ is chosen to connect to a new node. After the split operation, we obtain an updated graph $G_{i-1}$, as well as the split nodes $u^+$ and $v^+$, which are treated as a positive training example. Then, another node $v^-$, that is distinct from $u^+$ and $v^+$, is randomly chosen from the nodes in $\mathcal{V}_2^{G_{i-1}}$, and $(u^+, v^-)$ are viewed as a negative training example. The data tuple $(u^+, v^+, v^-, G_{i-1})$ is saved in the dataset $\mathcal{D}$. We also save the step count $n$ and the graph $G_0$ as "graph templates", which are later used to initialize G2SAT at inference time. The procedure is repeated $r$ times until the desired number of data points are gathered. Finally, G2SAT is trained with the dataset $\mathcal{D}$ to minimize the objective listed in Equation 3.

---

**Algorithm 1** G2SAT at training time

**Input:** Bipartite graphs $\mathcal{G}$, number of repetitions $r$
**Output:** Graph templates $\mathcal{T}$
$\mathcal{D} \leftarrow \varnothing, \mathcal{T} \leftarrow \varnothing$
**for** $k = 1, \ldots, r$ **do**
  $G \sim \mathcal{G}, n \leftarrow |\mathcal{E}^G| - |\mathcal{V}_2^G|, G_n \leftarrow G$
  **for** $i = n, \ldots, 1$ **do**
    $s \sim \{u | u \in \mathcal{V}_2^{G_i}, \text{Degree}(u) > 1\}$
    $(u^+, v^+, G_{i-1}) \leftarrow \text{NodeSplit}(s, G_i)$
    $v^- \sim \mathcal{V}_2^{G_{i-1}} \setminus \{u^+, v^+\}$
    $\mathcal{D} \leftarrow \mathcal{D} \cup \{(u^+, v^+, v^-, G_{i-1})\}$
  $\mathcal{T} \leftarrow \mathcal{T} \cup \{(G_0, n)\}$
Train G2SAT with $\mathcal{D}$ to minimize Eq. 3

---

**Algorithm 2** G2SAT at inference time

**Input:** Graph templates $\mathcal{T}$, number of output graphs $r$, number of proposed node pairs $o$
**Output:** Generated bipartite graphs $\mathcal{G}$
**for** $k = 1, \ldots, r$ **do**
  $(G_0, n) \sim \mathcal{T}$
  **for** $i = 0, \ldots, n - 1$ **do**
    $\mathcal{P} \leftarrow \varnothing$
    **for** $j = 1, \ldots, o$ **do**
      $u \sim \mathcal{V}_2^{G_i}, v \sim \{s | s \in \mathcal{V}_2^{G_i}, (s, x) \notin \mathcal{E}^{G_i}, (s, \neg x) \notin \mathcal{E}^{G_i}, \forall x \in N(u)\}$
      $\mathcal{P} = \mathcal{P} \cup \{(u, v)\}$
    $(u^+, v^+) \leftarrow \text{argmax}\{\mathbf{h}_u^T \mathbf{h}_v | (u, v) \in \mathcal{P}, \mathbf{h}_u = \text{GCN}(u), \mathbf{h}_v = \text{GCN}(v)\}$
    $G_{i+1} \leftarrow \text{NodeMerge}(u^+, v^+, G_i)$
  $\mathcal{G} = \mathcal{G} \cup \{G_n\}$

---

### 4.4 G2SAT at Inference Time

A trained G2SAT model can be used to generate graphs. We summarize the procedure in Algorithm 2. At graph generation time, we first initialize G2SAT with a graph template sampled from $\mathcal{T}$ gathered at training time, which specifies the initial graph $G_0$ and the number of generation steps $n$. Note that G2SAT can take bipartite graphs with arbitrary size as input and iterate for a variable number of steps. The reason we specify the initial state and the number of steps is to control the behavior of G2SAT and simplify the experiment setting.

At each generation step, we use the two-phase generation scheme described in Section 4.2. In the node proposal phase, we additionally make sure that the sampled node pair does not correspond to a

vacuous clause, i.e., if $u, v$ is a valid node pair, then $\forall x \in N(u)$, we ensure that $(v, x) \notin \mathcal{E}^{G_i}$ and $(v, \neg x) \notin \mathcal{E}^{G_i}$. We parallelize the algorithm by sampling $o$ random node pair proposals at once and feeding them to the node merging phase. In the node merging phase, although following Equation 2 would allow us to sample from the true distribution, we find in practice that it usually requires sampling a large number of candidate nodes pairs until a positive node pair is predicted by the GCN model. Therefore, we use a greedy algorithm that selects the most likely node pair to be merged among the $o$ proposed node pairs and merge those nodes. Admittedly, this biases the generator away from the true data distribution. However, our experiments reveal that the synthesized graphs are nonetheless reasonable. After $n$ steps, the generated graph $G_n$ is saved as an output.

## 5 Experiments

### 5.1 Dataset and Evaluation

**Dataset**. We use 10 small real-world SAT formulas from the SATLIB benchmark library [21] and past SAT competitions.[1] The two data sources contain SAT formulas generated from a variety of application domains, such as bounded model checking, planning, and cryptography. We use the standard SatElite preprocessor [11] to remove duplicate clauses and perform polynomial-time simplifications (for example, unit propagation). The preprocessed formulas contain 82 to 1122 variables and 327 to 4555 clauses.

We evaluate if the generated SAT formulas preserve the properties of the input training SAT formulas, as measured by graph statistics and SAT solver performance. We then investigate whether the generated SAT formulas can indeed help in designing better domain-specific SAT solvers.

**Graph statistics**. We focus on the graph statistics studied previously in the SAT literature [1, 34]. In particular, we consider the VIG, VCG and LCG representations of SAT formulas. We measure the modularity [33] (in VIG, VCG, LCG), average clustering coefficient [32] (in VIG) and the scale-free structure parameters as measured by variable $\alpha_v$ and clause $\alpha_c$ [1, 8] (in VCG).

**SAT solver performance**. We report the relative SAT solver performance, i.e., given $k$ SAT solvers, we rank them based on their performance over the SAT formulas used for training and the generated SAT formulas, and evaluate how well the two rankings align. Previous research has shown that SAT instances can be made hard using various post-processing approaches [13, 37, 42]. Therefore, the absolute hardness of the formulas is not a good indicator of how realistic the formulas are. On the other hand, it is not trivial for a post-processing procedure to precisely manipulate the relative performance of a set of SAT solvers. Therefore, we report the relative solver performance for a fairer comparison. We took the three best performing solvers from both the application track and the random track of the 2018 SAT competition [20], which are denoted as $I_1, I_2, I_3$, and $R_1, R_2, R_3$ respectively.[2] Our experiments confirm that solvers that are tailored to real-world SAT formulas $(I_1, I_2, I_3)$ indeed outperform solvers that focus on random SAT formulas $(R_1, R_2, R_3)$, over the training formulas. Therefore, we measure if on the generated formulas, the solvers $I$ similarly outperform the solvers $R$, as measured by ranking accuracy. All the run time performances are measured by wall clock time under carefully controlled experimental settings.

**Application: Developing better SAT solvers**. Finally, we consider the scenario where people wish to use the synthetic formulas for developing better SAT solvers. Specifically, we use either the 10 training SAT formulas or the generated SAT formulas to guide the hyperparameter selection of a popular SAT solver called Glucose [2]. We conduct a grid search over two of its hyperparameters — the variable decay $v_d$, that influences the ordering of the variables in the search tree, and the clause decay $c_d$, that influences which learned clauses are to be removed [12]. We sweep over the set $\{0.75, 0.85, 0.95\}$ for $v_d$, and the set $\{0.7, 0.8, 0.9, 0.99, 0.999\}$ for $c_d$. We measure the run time of the SAT solvers using the optimal hyperparameters found by grid search, over 22 real-world SAT formulas unobserved by any of the models. Since the number of training SAT formulas is limited, we expect that using the abundant generated SAT formulas will lead to better hyperparameter choices.

Table 1: Graph statistics of generated formulas (mean ± std. (relative error to training formulas)).

| Method | VIG | | VCG | | | LCG |
|---|---|---|---|---|---|---|
| | Clustering | Modularity | Variable $\alpha_v$ | Clause $\alpha_c$ | Modularity | Modularity |
| Training | 0.50±0.07 | 0.58±0.09 | 3.57±1.08 | 4.53±1.09 | 0.74±0.06 | 0.63±0.05 |
| CA | 0.33±0.08(34%) | 0.48±0.10(17%) | 6.30±1.53(76%) | N/A | 0.65±0.08(12%) | 0.53±0.05(16%) |
| PS(T=0) | 0.82±0.04(64%) | 0.72±0.13(24%) | 3.25±0.89(9%) | **4.70±1.59(4%)** | 0.86±0.05(16%) | **0.64±0.05(2%)** |
| PS(T=1.5) | 0.30±0.10(40%) | 0.14±0.03(76%) | 4.19±1.10(17%) | 6.86±1.65(51%) | 0.40±0.05(46%) | 0.41±0.05(35%) |
| G2SAT | **0.41±0.09(18%)** | **0.54±0.11(7%)** | **3.57±1.08(0%)** | 4.79±2.80(6%) | **0.68±0.07(8%)** | 0.67±0.03(6%) |

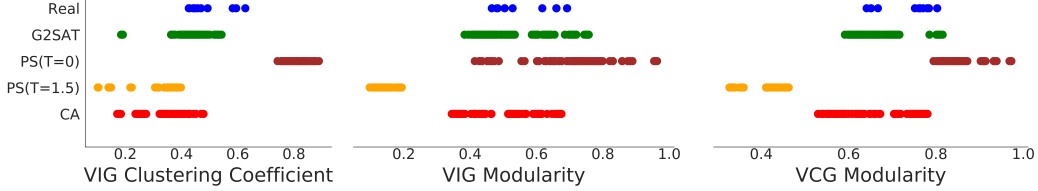

Figure 2: Scatter plots of distributions of selected properties of the generated formulas.

## 5.2 Models

We compare G2SAT with two state-of-the-art generators for real-world SAT formulas. Both generators are prescribed models designed to match a specific graph property. To properly generate formulas using these baselines, we set their arguments to match the corresponding statistics in the training set. We generate 200 formulas each using G2SAT and the baseline models.

**G2SAT**. We implement G2SAT with a 3-layer GraphSAGE model using mean pooling and ReLU activation [18] with hidden and output embedding size of 32. We use the Adam optimizer [25] with a learning rate of 0.001 to train the model until the validation accuracy plateaus.

**Community Attachment (CA)**. The CA model generates formulas to fit a desired VIG modularity $Q$ [15]. The output of the algorithm is a SAT formula with $n$ variables and $m$ clauses, each of length $k$, such that the optimal modularity for any $c$-partition of the VIG of the formula is approximately $Q$.

**Popularity-Similarity (PS)**. The PS model generates formulas to fit desired $\alpha_v$ and $\alpha_c$ [16]. The model accepts a temperature parameter $T$ that trades off the modularity and the $(\alpha_v, \alpha_c)$ measures of the generated formulas. We run PS with two temperature settings, $T = 0$ and $T = 1.5$.

## 5.3 Results

**Graph statistics**. The graph statistics of the generated SAT formulas are shown in Table 1. We observe that G2SAT is the only model that is able to closely fit *all* the graph properties that we measure, whereas the baseline models only fit some of the statistics and fail to perform well on the other statistics. Surprisingly, G2SAT fits the modularity even better than CA, which is tailored for fitting that statistic. We compute the relative error over the generated graph statistics with respect to the ground-truth statistics, and G2SAT can reduce the relative error by 24% on average compared with baseline methods. To further illustrate this performance gain, we plot the distribution of the selected properties over the generated formulas in Figure 2, where each dot corresponds to a graph. We see that G2SAT nicely interpolates and extrapolates on all the statistics of the input graphs, while the baselines only do well on some of the statistics.

**SAT solver performance**. As seen in Table 2, the ranking of solver performance over the formulas generated by G2SAT and CA align perfectly with their ranking over the training graphs. Both models are able to correctly generate formulas on which application-focused solvers $(I_1, I_2, I_3)$ outperform random-focused solvers $(R_1, R_2, R_3)$. By contrast, PS models do poorly at this task.

**Application: Developing better SAT solvers**. The run time gain of tuning solvers on synthetic formulas compared to tuning on a small set of real-world formulas is shown in Table 3. While all the generators are able to improve the SAT solver's performance by suggesting different hyperparameter

Table 2: Relative SAT Solver Performance on training as well as synthetic SAT formulas.

| Method | Solver ranking | Accuracy |
|---|---|---|
| Training | $I_2, I_3, I_1, R_2, R_3, R_1$ | 100% |
| CA | $I_2, I_3, I_1, R_2, R_3, R_1$ | **100%** |
| PS(T=0) | $R_3, I_3, R_2, I_2, I_1, R_1$ | 33% |
| PS(T=1.5) | $R_3, R_2, I_3, I_1, I_2, R_1$ | 33% |
| G2SAT | $I_1, I_2, I_3, R_2, R_3, R_1$ | **100%** |

Table 3: Performance gain when using generated SAT formulas to tune SAT solvers.

| Method | Best parameters | Runtime(s) | Gain |
|---|---|---|---|
| Training | (0.95, 0.9) | 2679 | N/A |
| CA | (0.75, 0.99) | 2617 | 2.31% |
| PS(T=0) | (0.75, 0.999) | 2668 | 0.41% |
| PS(T=1.5) | (0.95, 0.9) | 2677 | 0.07% |
| G2SAT | (0.95, 0.99) | **2190** | **18.25%** |

configurations, G2SAT is the only method that finds one that results in a large performance gain (18% faster run time) on unobserved SAT formulas. Although this experiment is limited in scale, the promising results indicate that G2SAT could open up opportunities for developing better SAT solvers, even in application domains where benchmarks are scarce.

## 5.4 Analysis of Results

**Scalability of G2SAT**. While existing deep graph generative models can only generate graphs with up to about 1,000 nodes [4, 17, 45], the novel design of the G2SAT framework enables the generation of graphs that are an order of magnitude larger. The largest graph we have generated has 39,578 nodes and 102,927 edges, which only took 489 seconds (data-processing time excluded) on a single GPU. Figure 3 shows the time-scaling behavior for both training (from 100k batches of node pairs) and formula generation. We found that G2SAT scales roughly linearly for both tasks with respect to the number of clauses.

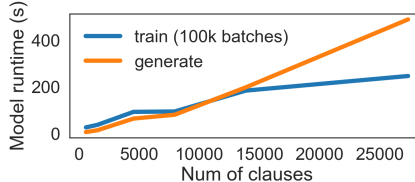

Figure 3: G2SAT Run time.

**Extrapolation ability of G2SAT**. To determine whether a trained model can learn to generate SAT instances different from those in the training set, we design an extrapolation experiment as follows. We train on 10 small formulas with 327 to 4,555 clauses, while forcing G2SAT to generate large formulas with 13,028 to 27,360 clauses. We found that G2SAT can generate large graphs whose characteristics are similar to those of the small training graphs, which shows that G2SAT has learned non-trivial properties of real-world SAT problems, and thus can extrapolate beyond the training set. Specifically, the VCG modularity of the large formulas generated by G2SAT is $0.81 \pm 0.03$, while the modularity of the small formulas used to train G2SAT is $0.74 \pm 0.06$.

**Ablation study**. Here we demonstrate that the expressive power of GCN model significantly affects the generated formulas. Figure 4 shows the effect of the number of layers in the GCN neural network model on the modularity of the generated formulas. As the number of layers increases, the average modularity of the generated formulas becomes closer to that of the training formulas, which indicates that machine learning contributes significantly to the efficacy of G2SAT. The other graph properties that we measured generally follow the same pattern as well.

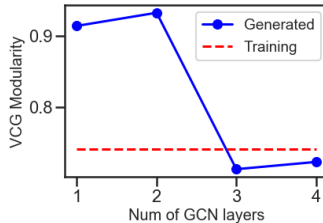

Figure 4: Ablation study.

## 6 Conclusions

In this paper, we introduced G2SAT, the first deep generative model for SAT formulas. In contrast to existing SAT generators, G2SAT does not rely on hand-crafted algorithms and is able to generate diverse SAT formulas similar to input formulas, as measured by many graph statistics and SAT solver performance. While future work is called for to generate larger and harder formulas, we believe our framework shows great potential for understanding and improving SAT solvers.

## Acknowledgements

Jure Leskovec is a Chan Zuckerberg Biohub investigator. We gratefully acknowledge the support of DARPA under No. FA865018C7880 (ASED) and MSC; NIH under No. U54EB020405 (Mobilize); ARO under No. 38796-Z8424103 (MURI); IARPA under No. 2017-17071900005 (HFC); NSF under No. OAC-1835598 (CINES) and HDR; Stanford Data Science Initiative, Chan Zuckerberg Biohub, Enlight Foundation, JD.com, Amazon, Boeing, Docomo, Huawei, Hitachi, Observe, Siemens, UST Global. The U.S. Government is authorized to reproduce and distribute reprints for Governmental purposes notwithstanding any copyright notation thereon. Any opinions, findings, and conclusions or recommendations expressed in this material are those of the authors and do not necessarily reflect the views, policies, or endorsements, either expressed or implied, of DARPA, NIH, ONR, or the U.S. Government.

## Footnotes

*The two first authors made equal contributions.

[1]Link to code and datasets: `http://snap.stanford.edu/g2sat/`

[2]Any SAT formula can be converted to an equisatisfiable CNF formula in linear time [39].

[1]http://www.satcompetition.org/

[2]The solvers are, in order, MapleLCMDistChronoBT, Maple_LCM_Scavel_fix2, Maple_CM, Sparrow2Riss-2018, gluHack, glucose-3.0_PADC_10_NoDRUP [20].

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
