[Supplementary Material]

# Appendix for "G2SAT: Learning to Generate SAT Formulas"

## 1 Implementation Details

### 1.1 Removing Trivial Components of SAT formulas Generated by PS

We found that almost all formulas generated by the PS model were trivially unsatisfiable when it is tasked with generated formulas with similar metrics as formulas in the traning set. In order to better demonstrate the SAT solver behaviors on the PS model, we increased the difficulty of its generated formulas by performing a lightweight post-processing step that iteratively removes clauses leading to short unsatisfiable proof. Concretely, for each generated formula, we used a SAT solver (Picosat [2]) to try to solve it within a given conflict[1] budget. If the solver were able to prove that the formula is unsatisfiable within that budget, we removed a clause from the unsatisfiable core (i.e., a subset of all clauses that is unsatisfiable) that Picosat returned and repeated the same process on the pruned formula, until it became satisfiable or no longer solvable within the conflict budget. We found that this efficient approach, only removes a low proportion of clauses and has small impact on the graph theoretic properties, but significantly increases the difficulty of the formulas generated by PS.

## 2 Experiment Details

### 2.1 Computing facilities

The G2SAT model is trained on a single NVIDIA RTX-2080Ti GPU. We evaluate the performance of SAT solvers on a cluster equipped with Intel Xeon E5-2637 v4 CPUs running Ubuntu 16.04 and we dedicated 2 cores, 8000 MB RAM for each job. For each formula, we gave each solver a 10 minutes timeout.

### 2.2 Evaluation

We used an implementation of the Louvain Algorithm to measure the modularities [3].

We used an implementation of the maximum likelihood method for computing an estimate of $\alpha_v$ and $\alpha_c$ [4, 1].

### 2.3 More details on baseline methods

**Community Attachment (CA)**. The CA model generates formulas to fit a desired VIG modularity [5]. The model takes in five inputs $n, m, k, c, Q$, where $n$ is the number of variables, $m$ the number of clauses, $k$ the length of each clause, $c$ the size of a partition of the VIG, and $Q$ is the desired VIG modularity. The output of the algorithm is a SAT formula with $n$ variables and $m$ clauses,

each of length $k$, such that the optimal modularity for any $c$-partition of the VIG of the formula is approximately $Q$.

**Popularity-Similarity (PS).**

The PS model generates formulas to fit desired $\alpha_v$ and $\alpha_c$ [6]. In addition, the formulas generated by PS are guaranteed to have high modularity. The model takes in seven inputs $n, m, k, K, \alpha_v, \alpha_c, T$, where $n, m$ are the same with CA, $k$ the minimum clause length, $K$ the average clause length, and $T$ a hyper-parameter that decides the trade-off between modularity and $\alpha_v, \alpha_c$. We use two versions of PS, with $T = 0$ and $T = 1.5$.

## Footnotes

[1] A SAT solver finds a conflict when it finds a partial assignment of values to variables that cannot satisfy the formula.