[Reviews · NeurIPS 2019]

Reviewer 1



In this paper, a deep generative model that learns to generate SAT formulae similar to a set of formulae provided as input for the training time is proposed. The approach is novel as far as I can tell. The paper is well written and provides sufficient level of details. The experimental evaluation is brief, but shows promise. indeed, G2SAT seems to generate CNFs which are similar to those provided as input, and which can be used as additional benchmarks. In the training, 10 smallest SATLIB/SAT competitions benchmarks are used, resulting in CNFs with 82 to 1122 variables and 327 to 4555 clauses. Would G2SAT scale to using larger instances? In the generation of new instances, are these of similar size to the instances used in the training time? I did not find a mention about this. Furthermore, it would be interesting to know about the the time wise scaling of G2SAT (this is essential to demonstrate the stated efficiency of G2SAT). How long does it take to generate new instances? Does this scale to larger instances? Pros: - A way to generate SAT formulas with similar properties as the formulas provided as input the the training of the model - Experiments suggest that the instances generated indeed resemble the original ones provided as input in terms of several measures Cons: - No mention of how does the approach scale - No access to code during reviewing ======================== After response period: The author's response clarified many of the concerns raised in the reviews. The additional results provided should definitely be included in the paper.

Reviewer 2



The procedure for generating SAT instances is original as far as the reviewer knows (although the reviewer is not very familiar with related literature). The proposed method seems technically sound. The empirical evaluation seems fair: the results are compared to several relevant baselines from the literature and the generated SAT instances seems to align better with the data they are trying to mimic on most of the metrics, including the ranking of industrial vs random SAT solver performance. It would be nice to see actual run-time numbers for the SAT solvers - even though the argument in lines 263-264 suggests that post-processing can significantly change these numbers, the reviewer thinks that the results should be still useful to the reader. The application of tuning SAT solver hyperparameters seems useful. It is unclear how much learning contributes to the success of the method - comparison of variations of the method with a higher or lower capacity GCN would be useful to gauge how much learning occurs on the rather small dataset of SAT instances. The paper is clearly written, although the reviewer didn't find exact description of the networks and optimization procedures used for training in the submission. The reviewer thinks that the paper explores an interesting problem at the intersection of SAT solving and machine learning, and future work is likely to build on the ideas presented in the paper.

Reviewer 3



That these types of techniques are really essential for "developing better and faster SAT solvers" is quite questionable, since the SAT community has been making quite good progress in pushing solvers forward and still are, and the benchmark situation has considerably improved over the years. Nevertheless, from an academic perspective building generators for real-world like formulas in interesting. Perhaps the biggest issue with the work is in that it does not really seem to scale. The authors use the 10 *smallest* "real-world" instances in their experiments. Small instances are really not interesting from the perspective of SAT solver development, as most real-world instances are quite huge. In my opinion the authors would need to address at the very least the scalability issues to warrant publication at a major venue. Regarding the comparison of graph statistics of the generated instances, I do not see what to make of these. It is unclear to me to what extent the instances are *different* from the instances started from. In fact, one could also interpret the fact that solvers rankings do not change as being an artifact of the potential fact that the instances do not really change much. The authors should convincingly argue that the generated instances would be in some sense really (in an interesting way) also *different* from the original instances (slightly confusing a formula can be done without any machine learning, most definitely!) The "application" of "developing better SAT solvers" seems somewhat absurd to me, as I do not understand what we could really take away from the presented observations, and what is the actual supposed role of the generation technique presented to this. Finally there appears to be very closely related work presented at SoCS'19: https://aaai.org/ocs/index.php/SOCS/SOCS19/paper/view/18390 The authors should cite this work and explain their own contributions in relations to that work.

[Author Response · NeurIPS 2019]

We thank the reviewers for their positive feedback and thoughtful suggestions. Overall, the reviewers found this research
worthwhile and interesting. R2 and R4 requested further demonstration of the scalability of G2SAT. R3 asked for more
ablation studies of our techniques. R4 raised concerns regarding our main motivation, questioned the extrapolation
ability of G2SAT, and pointed out some related work. We will carefully revise the final paper to clarify these points.

**1. Motivation (R4)**. R4 questioned the contribution of our technique to the SAT community, thus we clarify as follows.
**First**, G2SAT manifests an efficient and general technique to generate interesting SAT benchmarks. While we agree
with R4 that "*the benchmark situation has improved over the years*", new interesting benchmarks are still demanded and
highly welcomed by the SAT community. For example, new benchmarks, both real and synthetic, are called for during
each year's SAT competition. **Second**, G2SAT demonstrates a novel data-driven approach to improve SAT-solving.
Although, as R4 correctly pointed out, significant progress in SAT solvers has been made over the past few years, we
point out that those improvements result mainly from better hand-crafted heuristics and software engineering. On the
other hand, the promising result of an experiment in our paper (line 275-284), where we showed that G2SAT formulas
can be used to better tune the hyper-parameters of a SAT solver, suggests the exciting opportunities to improve SAT
solvers in a data-driven manner. This direction depends on having a large number of realistic formulas, and we propose
G2SAT as one way to obtain such formulas. Therefore, we believe our techniques, in addition to being theoretically
interesting, are meaningful to the SAT community. We will be more specific about these points in the revised version.

**2. Experiment (R2, R3, R4)**. We conduct additional experiments to address the reviewers' concerns.

**Scalability of G2SAT (R2, R4)**. It is worth noting that existing deep graph generative models can only generate
relatively small graphs, ranging from tens of nodes (GraphVAE), hundreds
of nodes (Learning Deep Generative Model of Graphs, GCPN) up to 1,000
nodes (GraphRNN, Graphite, NetGAN). In contrast, the novel design of the
G2SAT framework (elaborated on in Section 4.2) enables the generation of
graphs an order of magnitude larger than those in previous work. Notably,
in our additional experiments, the largest graph we generate has 39,578
nodes and 102,927 edges, which only took 489 seconds (data-processing
time excluded) to generate on a single GPU. Figure 1 further shows the

Figure 1: Run time scaling of G2SAT

time-scaling for both training (from 100k batches of node pairs) and formula generation. We found that G2SAT scales
roughly linearly for both tasks with respect to the number of clauses.

**Extrapolation ability of G2SAT (R4)**. To address the concern of R4 on whether a trained model can learn to generate
SAT instances different from those in the training set, we design an extrapolation
experiment as follows. We train on 10 small formulas with 327 to 4,555 clauses,
while forcing G2SAT to generate large formulas with 13,028 to 27,360 clauses. Note
that none of the baseline methods can accomplish the same task, as they can only
mimic a given SAT formula. On the contrary, G2SAT can generate large graphs whose
properties are similar to those of the small training graphs, which shows that G2SAT
has learned non-trivial properties of real-world SAT problems, and thus can extrapolate
beyond the training set. Specifically, the VCG modularity of the large generated
formulas is $0.81 \pm 0.03$, while the modularity of the training formulas is $0.74 \pm 0.06$.

Figure 2: Ablation Study

**Ablation study (R3)**. Figure 2 shows the effect of the number of layers of the GCN neural network model on the
modularity of the generated formulas. As the number of layers increases, the average modularity of the generated
formulas becomes closer to the training formulas, which indicates that machine learning contributes significantly to the
efficacy of G2SAT. The other graph properties that we measured generally follow the same pattern.

**3. Related work (R4)**. We thank R4 for pointing out the related work, a 2-page extended abstract that came out on July
5, which is more than 1 month after the NeurIPS deadline. Our paper differs from that work in at least 4 ways. (1)
G2SAT is over bijective LCG representation of SAT, rather than the LIG representation that has ambiguity. (2) G2SAT
proposes a novel and scalable bipartite graph generator, while that paper applies an existing model (NetGAN) that can
only perturb one given SAT instance. (3) We show that G2SAT-generated formulas resemble the training formulas
in SAT-solver performance. (4) We investigate how a SAT generator can potentially help design better SAT solvers.
Nevertheless, we recognize the relation of that work to our paper, and will cite that work in the revised version.

**4. Code and implementation details (R2, R3)**. We promise to open-source our code if our paper is accepted. Due to
page limits, we did not describe the full details of the model. In summary, we use a standard 3-layer GCN with 32
hidden dimensions in each layer and ReLU activation. We train our model using Adam optimizer with learning rate
0.001, over 10M batches of node pairs for all experiments. We will include these details in the revised version. As
requested by R3, we will report the actual total runtime of SAT-solvers on the generated formulas in the revised version.

[Meta-Review · NeurIPS 2019]

Dear authors: your paper was considered in great detail by the reviewers, and their evaluation of the paper was quite positive. Also, the rebuttal clarified various aspects of the paper. The topic at hand, generating SAT formulas using generative modelling, is quite interesting and the use of sophisticated ML tools seems quite promising. Your paper will be accepted, but I would strongly encourage you to consider the reviewers' detailed comments when finalizing your camera-ready document.